# Maximal Walking Distance in Persons with a Lower Limb Amputation

**DOI:** 10.3390/s20236770

**Published:** 2020-11-26

**Authors:** Cheriel J. Hofstad, Kim T.J. Bongers, Mark Didden, René F. van Ee, Noël L.W. Keijsers

**Affiliations:** 1Department of Research & Innovation, Sint Maartenskliniek, 6500 GM Nijmegen, The Netherlands; k.bongers@interzorgthuiszorg.nl (K.T.J.B.); m.didden@tolbrug.nl (M.D.); n.keijsers@maartenskliniek.nl (N.L.W.K.); 2Department of Rehabilitation, Tolbrug, 5223 GZ Den Bosch, The Netherlands; 3Department of Rehabilitation, Sint Maartenskliniek, 6500 GM Nijmegen, The Netherlands; r.vanee@maartenskliniek.nl; 4Department of Rehabilitation, Donders Institute for Brain, Cognition and Behaviour, Radboud University Medical Center, 6525 AJ Nijmegen, The Netherlands

**Keywords:** inertial measurement units, walking distance, lower limb amputation, rehabilitation, gait

## Abstract

The distance one can walk at a time could be considered an important functional outcome in people with a lower limb amputation. In clinical practice, walking distance in daily life is based on self-report (SIGAM mobility grade (Special Interest Group in Amputee Medicine)), which is known to overestimate physical activity. The aim of this study was to assess the number of consecutive steps and walking bouts in persons with a lower limb amputation, using an accelerometer sensor. The number of consecutive steps was related to their SIGAM mobility grade and to the consecutive steps of age-matched controls in daily life. Twenty subjects with a lower limb amputation and ten age-matched controls participated in the experiment for two consecutive days, in their own environment. Maximal number of consecutive steps and walking bouts were obtained by two accelerometers in the left and right trouser pocket, and one accelerometer on the sternum. In addition, the SIGAM mobility grade was determined and the 10 m walking test (10 MWT) was performed. The maximal number of consecutive steps and walking bouts were significantly smaller in persons with a lower limb amputation, compared to the control group (*p* < 0.001). Only 4 of the 20 persons with a lower limb amputation had a maximal number of consecutive steps in the range of the control group. Although the maximal covered distance was moderately correlated with the SIGAM mobility grade in participants with an amputation (*r* = 0.61), for 6 of them, the SIGAM mobility grade did not match with the maximal covered distance. The current study indicated that mobility was highly affected in most persons with an amputation and that the SIGAM mobility grade did not reflect what persons with a lower limb amputation actually do in daily life. Therefore, objective assessment of the maximal number of consecutive steps of maximal covered distance is recommended for clinical treatment.

## 1. Introduction

The two primary concerns for people with a lower limb amputation are mobility [1,2] and wearing comfort of the prosthesis, in which mobility is most relevant for their quality of life [3,4]. However, many persons with a lower limb amputation report that they are unable to use their prosthesis to the extent they desire [2] and, moreover, they lose their independence [5,6]. To function independently, one should be able to walk sufficient bouts. Therefore, in the context of independency, the walking distance one can walk consecutively could be considered as an important outcome in persons with a lower limb amputation. In clinical practice, the self-reporting SIGAM (Special Interest Group in Amputee Medicine) mobility grades [4] are often used to classify prosthetic users. The SIGAM mobility grades describe a single-item scale comprising six clinical grades (A–F) of amputee mobility, and the scale consists of 21 ‘yes’/‘no’ items. The SIGAM mobility grades include a walking distance item; a threshold of 50 m at a time is used as a benchmark to denote an improvement of mobility [4] and reflects sufficient independency. It is known, however, that people tend to overestimate their physical activity when self-report measures are used [7,8,9]. As the SIGAM mobility grade is a self-reporting questionnaire, it is very likely that the activity levels are overestimated in the SIGAM mobility grades. This results in false positive outcomes, also known as bias towards independency. Since clinical interventions, like prosthetic fitting, are partially based on questionnaires assessing functional level [10], it is conceivable that clinical care might be subject to bias or subjectivity.

In contrast to self-reported measures (diaries and questionnaires [11,12,13,14]), there are also technical approaches that were used to assess prosthetic mobility. All techniques differ in the type and number of mobility aspects they measure, ranging from categories of ambulation to prosthetic use over a variety of ambulation activities [12] and performance tests in laboratory settings [15]. Another more objective way to measure mobility is the use of activity monitors [16,17,18,19,20,21,22,23,24]. The advantage of activity monitors is that they can measure long-term and continuously in a person’s own environment, and assess what persons with a lower limb amputation actually do, in a reliable and valid way [25]. Although it was demonstrated that persons with a lower limb amputation are significantly less physically active compared to the age-matched controls [17,18], none of the studies focused on the length of walking bouts and the number of consecutive steps in these bouts. 

The aim of this study was to assess the number of consecutive steps and walking bouts in persons with a lower limb amputation and age-matched controls in daily life, using an accelerometer sensor. We hypothesized that the maximal number of consecutive steps was correlated to the level of the SIGAM mobility grades. We were particularly interested in whether physically active or independent persons with a lower limb amputation (SIGAM mobility grade D or higher) covered longer distances than 50 m during walking bouts, which is an important benchmark for mobility, as stated by Ryall [4]. We also assessed the relationship between the SIGAM mobility grade, maximal covered distance and preferred walking velocity, to indicate the effect of gait capacity on physical functioning. Age-matched subjects were included for comparison.

## 2. Materials and Methods

### 2.1. Subjects

Patients were recruited from the Prosthetics and Orthotics Centre in Nijmegen and from the prosthetic training group at the rehabilitation clinic Sint Maartenskliniek in Nijmegen, The Netherlands. Persons with a lower limb amputation were included when they had a unilateral transfemoral or transtibial amputation or knee exarticulation, were at least 18 years old, and had no cognitive disorders. They had to be free from neurological and clinical orthopedic problems (other than the amputation), without stump pain, stump wounds, and foot pathology, which could influence their daily activities. A control group of age-matched subjects without an amputation also participated in this study. All participants gave written informed consent in accordance with the Declaration of Helsinki. The study was approved by the internal review board of the Sint Maartenskliniek. The study was carried out in the Netherlands, in accordance with the applicable rules concerning the review of research ethics committees, and did not fall within the remit of Medical Research Involving Human Subjects Act.

### 2.2. Accelerometers

The accelerometers (62 mm (length) × 41 mm (width) × 18 mm (height)) used in this study were tri-axial piezo-capacitive MiniMods from Dynaport (McRoberts BV, The Hague, The Netherlands). The sample rate of the accelerometers was 100 Hz and data were stored on secure digital (SD) memory cards. Three accelerometers were used during the measurement. Two were placed in the left and right trouser pocket and one on the lower part of the sternum of the subject. The accelerometer that was placed on the sternum was attached with means of a 10 cm wide elastic band around the chest, to prevent irritation of the skin.

### 2.3. Protocol

The accelerometers were worn over two consecutive days in the participant’s own environment. The researcher explained the measurement protocol, instructed the patient on how to attach and detach the accelerometers and administered the SIGAM mobility grade. The participants were instructed to perform their normal daily life activities during the two measurement days. At the end of both measurement days, the participant had to fill out a short questionnaire on whether the activities performed were representative for someone’s usual daily activities. The accelerometers were not worn overnight.

To be able to calculate the maximal walking distance, step length was estimated by data form a 10-m walk test (10 MWT). In addition, the 10 MWT was also used to assess the preferred walking velocity, which is an excellent indicator of gait capacity. After attaching the accelerometers on the first day, the participant performed a 10 MWT. The participant was instructed to walk 10 m at his own comfortable pace. The start and finish of the 10 MWT were marked in the accelerometer data by pushing a remote button, which was connected to the accelerometers. The researcher timed the 10 MWT and counted the number of steps. 

### 2.4. Data Analysis and Outcome Measures

The main outcome measure of this study was the number of steps a subject walked consecutively during the two measuring days. Walking could be well detected by accelerometers on a thigh [26] or trunk [27]. Two custom written algorithms were used (MATLAB 7.1, The Mathworks Inc, Natick, MA, USA). The first algorithm identified walking bouts, in which a subject was walking. A subject was considered as walking when the orientation of all three accelerometers was upright and there was sufficient movement of the sensors. As a measure for the movement of the sensor, we took the square root of the sum of squares of the derivative of the three orthogonal accelerometer signals [28]. Finally, the signals of the accelerometers should have a repetitive character, which was determined by the autocorrelation of the accelerometer signals. The second algorithm counted the number of consecutive steps within each walking bout. The number of steps for each walking bout was calculated by dividing the time of the walking bout by the step frequency of the walking bout, which was the dominant frequency in the auto correlation of the accelerometer signals. Subsequent walking bouts with an interval within 1 s were seen as a single walking bout.

All walking bouts were visually checked on time and steps, and the remaining data were visually screened for walking bouts missed by the algorithm. Walking bouts missed by the algorithm were added.

In addition to the number of consecutive steps within each walking bout, we were interested in the frequency of walking bouts per hour. Therefore, categories of walking bouts were created in bins of 5 steps (for the walking bouts in which 0 to 50 consecutive steps were walked), bins of 25 steps (from 50–100 consecutive steps), and bins of 100 steps (from 100–400 consecutive steps). These frequencies were determined for both the persons with a lower limb amputation and the elderly control group.

To estimate the maximal walking distance in the persons with a lower limb amputation, the maximal number of consecutive steps was multiplied by the individual step length, based on the 10 MWT. The individual step length was 10 m divided by the number of steps needed to accomplish the 10 MWT. This estimated maximal walking distance was compared with the specific answer on the walking distance questions of the SIGAM mobility grades (“Do you usually manage to walk more than 50 m (55 yards) at a time?”).

### 2.5. Statistical Analysis

Differences in the group characteristics and results of the 10 MWT were calculated with a nonparametric independent samples test (Mann–Whitney test). To calculate the difference of the frequency per hour between the groups (persons with a lower limb amputation vs. the elderly control group) and walking bouts, a mixed model ANOVA was performed with persons with a lower limb amputation or the elderly control group as between-factor, and walking bout bins as the within-group factor. Spearman’s rank correlation coefficients were calculated between 10 MWT, the SIGAM mobility grade, and the maximal covered walking distance, to indicate the relationship between gait capacity and physical functioning. Statistics were performed in SPSS 12.0.1 (SPSS Inc. Chicago, IL, USA). Differences were considered significant when *p* < 0.05.

## 3. Results

### 3.1. Participants

Twenty subjects with a lower limb amputation and ten age-matched controls participated in this study. See Table 1 for characteristics of both groups. Nineteen subjects had data on two complete measurement days. Eleven subjects generated data on only one complete day, because some participants failed to start or recharge the accelerometers adequately or due to technical problems. Mean measurement time for the complete days was 9:45 h ± 2:37 (SD) for the persons with a lower limb amputation and 11:20 h ± 1:40 (SD) for the elderly control group. All subjects, except one control subject who was sick during the measurement days, indicated that the measurement days were normal with regards to their standard daily activities.

### 3.2. Maximal Number of Consecutive Steps and 10 MWT

Table 2 shows the median and interquartile range of the maximal number of consecutive steps and the 10 MWT. For both the maximal consecutive steps and the 10 MWT, the elderly control group performed better than the persons with a lower limb amputation (*p* < 0.001 for the Mann–Whitney test).

#### Maximal Number of Consecutive Steps per Individual

The maximal number of consecutive steps was significantly larger in the elderly control group (*p* < 0.001, Table 2). However, some active persons with a lower limb amputation achieved similar maximal consecutive steps. Figure 1 shows the maximal number of consecutive steps for each subject. All elderly controls achieved more than 250 consecutive steps, except one. This elderly control subject reached a maximal of 94 steps, but reported on the activities questionnaire that she walked less than normal, due to illness. In contrast, only 4 of the 20 persons with a lower limb amputation achieved the 250 consecutive steps. Furthermore, eight persons with a lower limb amputation had even less than 100 consecutive steps. It was remarkable that one person with a lower limb amputation (SIGAM mobility grade F) revealed the highest maximal number of consecutive steps of almost 2500.

### 3.3. Frequency of Number of Steps per Hour

Figure 2 shows the frequency per hour per bin (number of consecutive steps). The mixed ANOVA revealed an interaction effect (F14,392 = 2.41, *p* = 0.003) and a significant main effect for the number of consecutive steps (F14,29 = 56.1, *p* < 0.001) and no significant main effect for group (F1,28 = 4.13, *p* = 0.052). Post-hoc analysis showed that the elderly controls had significantly more walking bouts with 10–25 consecutive steps and more than 100 consecutive steps (as indicated by the * in Figure 2).

### 3.4. 10 MWT and Maximal Covered Walking Distance 

The left panel of Figure 3 shows the performance on the 10 MWT for the SIGAM mobility grades for all persons with a lower limb amputation and the elderly controls (EC). Spearman’s rank correlation between the SIGAM mobility grade and the 10 MWT was −0.78 (*p* = 0.0001). Based on the 10 MWT, the median estimated maximal covered distance in the persons with a lower limb amputation was 67 m (with an interquartile range of 22–93). The maximal covered distance is shown for the SIGAM mobility grades in the center panel of Figure 3. Obviously, the higher the SIGAM mobility grade, the higher the maximal covered distance (*r* = 0.61, *p* = 0.006). Nevertheless, a closer look showed that even some persons with a lower limb amputation with a SIGAM mobility grade higher than C did not reach the 50 m. The maximal covered distance was also significantly correlated with the 10 MWT (*r* = −0.66, *p* = 0.002).

## 4. Discussion

The goal of this study was to assess the maximal covered walking distance and walking bouts in a wide range of persons with a lower limb amputation in daily life. Forty percent of the persons with a lower limb amputation (8 out of 20) did not reach walking distances of 50 m during a single walking bout, which was indicated as an important benchmark for mobility and, therefore, important for independent living and social participation. There was a significant positive correlation between the maximal covered distance and the SIGAM mobility grades (Figure 3). In contrast to the persons with a lower limb amputation, the elderly control group, except for the sick subject, covered a walking distance of at least 150 m, based on the maximal number of consecutive steps of at least 300 (Figure 1). These results imply that the current SIGAM mobility grades do not sufficiently reflect what a person with lower limb amputation actually does in daily life, but more what a person is able to do.

Several studies performed activity measurements in persons with a lower limb amputation with daily duration of dynamic activities or daily number of steps as the main outcome measure [16,17,18,19,20,21,22,23,24]. The lower number of walking bouts, especially in the long walking bouts, compared to the age-matched control subjects, supports the finding that persons with a lower limb amputation are less active. However, none of these studies investigated walking bouts and the related maximal number of consecutive steps. For persons with a lower limb amputation, maximal walking distance is an important measure for social mobility and ADL independence. Since SIGAM mobility grades uses the 50 m walking distance as a limit for indoor and outdoor walking, this 50 m limit should correspond with independence, and the level at which a person can participate in society [4]. Forty percent of the persons with a lower limb amputation did not cover a walking distance of more than 50 m. Except for the sick subjects, the elderly control group had a maximal number of steps of at least 300, which was at least 150 m with a 0.5 m step length. Therefore, walking bouts of at least 300 steps seemed to be the lower bound for walking mobility in the elderly control. In contrast, only 4 of 12 persons with a lower limb amputation with normal or near normal gait (3 persons with a lower limb amputation with SIGAM mobility scale grade F and one person with a lower limb amputation with grade D) took more than 300 steps consecutively. A minimal walking distance of approximately 300–350 m is required for community walking tasks, such as walking from the parking lot to the grocery shop or visiting a health care practitioner [29,30,31]. In our study, 4 out of 20 persons with a lower limb amputation and 7 out of the 10 elderly control had walking bouts of more than 600 steps, which indicated at least community walking. Hence, walking mobility was affected in most persons with a lower limb amputation who were defined as normal or near normal walkers.

The limited walking distance at a time, for persons with a lower limb amputation, could be compensated by walking consecutive short distances more frequently, with rest periods in between. However, persons with a lower limb amputation had significantly smaller short walking bouts compared to the elderly control. Furthermore, detailed analysis revealed that in the persons with a lower limb amputation data, consecutive short walking bouts with rest periods were not present, making it impossible to reach similar long walking distances as the elderly control. There might be several reasons why persons with a lower limb amputation avoid walking long distances. One explanation might be that the persons with a lower limb amputation adapt their walking distance to keep their heart rate response within a normal range [18]. Another explanation might be that persons with a lower limb amputation had a poorer joint coordination, and thus might be easier to get fatigued, feel discomfort, and have an unstable gait [32,33]. It seems that walking is already a maximum effort for a great part of the persons with a lower limb amputation. Beside the physical limitation, outdoor gait performance of the persons with a lower limb amputation is of course also dependent on a variety of other factors, including personal interest, weather, terrain, comorbidities, prosthetic fit, social interactions, etc. [24].

Evaluation of daily functioning of persons with a lower limb amputation is highly based on questionnaires such as the SIGAM mobility grade, which are based on self-report and estimates of the physician. Several studies found that one of the risks of self-report activity questionnaires is an overestimation of activity levels when using self-reported measures [7,8,9]. Bootsma-van der Wiel et al. [34] found that discrepancies between what the elderly (>85 years) can do and actually do in activities of daily living had important consequences when estimating disability in old people. As a consequence, incorrect assessment of daily functioning might influence the care given. The clear positive correlation between maximal covered distance and the SIGAM mobility grades and maximal covered distance and 10 MWT implies a high association between gait mobility and gait capacity, which justifies the SIGAM mobility grade as an evaluation for daily functioning. However, the limit of 50 m walking at a time as a threshold for the SIGAM mobility grades of D and higher was not established by all persons with a lower limb amputation, with a SIGAM mobility of D or higher. Furthermore, 2 of the 6 persons with a lower limb amputation with a SIGAM mobility grade of B or C, covered a larger distance than 50 m. Therefore, a discrepancy exists between the SIGAM mobility grades and performance in daily life, which corresponds to the results of Albert et al. [35]. This finding implies that the SIGAM mobility grade of persons with a lower limb amputation is more dependent on the type of activities one can perform, than purely on walking distance. Therefore, daily functioning should not only question what a person with a lower limb amputation can do but should also monitor the amputees actual daily activities and walking distance. For the assessment of daily functioning, more information can be obtained than only maximal number of consecutive steps and gait bouts. For example, the distribution of walking bouts across the day, use of walking aids, how long the prosthesis was worn during the day, and specific activities for a person with a lower limb amputation.

### Limitations

A limitation of the current study is that data collection was limited to two days and the maximum covered distance was estimated by multiplying the number of steps, with the step length measured with the 10 MWT. Although a larger number of measurement days than 2 would have resulted in a more accurate estimates of the maximal number of consecutive steps, the walking bouts of at least 300 steps in the elderly control group indicated that 1–2 days was sufficient to indicate their mobility. The estimated covered distance was most likely an overestimate since daily life walking was less regular than walking during a 10 MWT. Inertial measurement units attached to the shoe or ankle would be a better alternative as it estimates gait velocity and step length in a valid and reliable way [36,37]. We chose the most convenient and easy way, by focusing on the number of steps, which could also be simply assessed by using, for example, a smart watch or a smart phone [38,39,40,41]. The relatively small sample size did not allow us to perform a sub-analysis within the persons with a lower limb amputation. We expect that the level of amputation and reason for amputation group would affect the maximal covered distance. Persons with a transfemoral amputation would most likely have a reduced walking distance compared to the persons with a transtibial amputation.

## 5. Conclusions

The current study indicates that mobility is highly affected in most persons with a lower limb amputation and that the SIGAM mobility grade does not reflect what persons with a lower limb amputation actually do in daily life. Therefore, objective assessment of the maximal number of consecutive steps of the maximal covered distance, is recommended for clinical treatment.

## Figures and Tables

**Figure 1 sensors-20-06770-f001:**
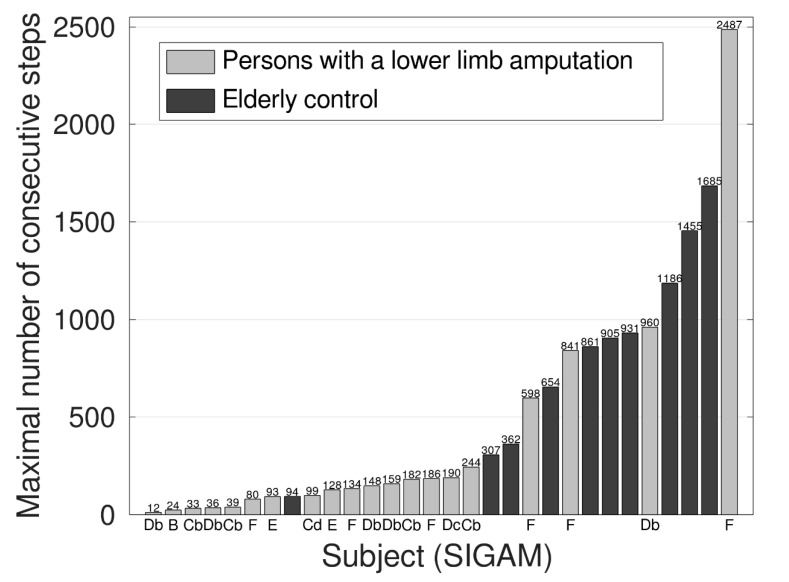
Maximal numbers of consecutive steps of each subject. The SIGAM mobility grade is given for every subject of the persons with a lower limb amputation. B = Therapeutic use only for transfers, C = Walks on level ground less than or equal to 50 m with (Cb)/without aids (Cd), D = Walks outdoor on level ground only, in good weather, more than 50 m with 2 crutches/sticks (Db) or 1 crutch/stick (Dc), and E = Walks more than 50 m, no aids, except in adverse terrain or weather, F = normal or near normal gait.

**Figure 2 sensors-20-06770-f002:**
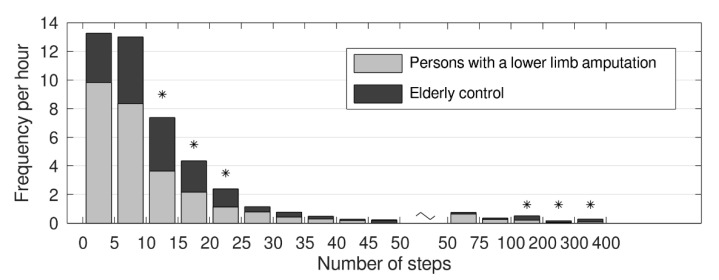
The mean frequency per hour of the number of steps per bin. Grey bars are persons with a lower limb amputation, black bars are the elderly controls. * Post-hoc difference between the persons with a lower limb amputation and the elderly control group.

**Figure 3 sensors-20-06770-f003:**
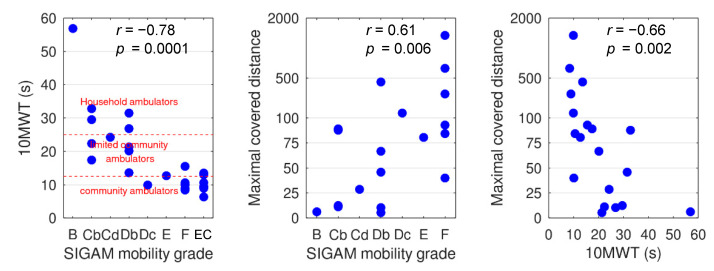
Relation between the SIGAM mobility grades, the 10 MWT, and the maximal covered distance. B = Therapeutic use only for transfers, C = Walks on level ground less than or equal to 50 m with (Cb)/without aids (Cd), D = Walks outdoor on level ground only, in good weather, more than 50 m with 2 crutches/sticks (Db) or 1 crutch/stick (Dc), and E = Walks more than 50 m, no aids, except in adverse terrain or weather, F = normal or near normal gait. EC = elderly controls.

**Table 1 sensors-20-06770-t001:** Characteristics of persons with a lower limb amputation and elderly controls, median (interquartile range).

Characteristic	Persons with A Lower Limb Amputation	Elderly Control Group	*p*-Value
Gender (M:F)	13:7	5:5	
Age (years)	68 (60–74)	76 (69–81)	0.43
Height (cm)	171 (165–179)	172 (168–175)	0.93
Weight (kg)	77 (67–85) *	78 (75–78)	0.83
Amputation level	TT *n* = 9 KE *n* = 4 TF *n* = 7	n.a.	
Reason for amputation	Traumatic *n* = 6 Vascular *n* = 10 Oncological *n* = 2 VOther *n* = 2	n.a.	
SIGAM mobility grade	B *n* = 1 C *n* = 5 D *n* = 6 E *n* = 2 F *n* = 6	n.a.	

TT = Transtibial amputation; KE = Knee exarticulation; TF = Transfemoral amputation; SIGAM mobility grades: B = Therapeutic use only for transfers, C = Walks on level ground less than or equal to 50 m with/without aids, D = Walks outdoor on level ground only, in good weather, more than 50 m with/without walking aid, and E = Walks more than 50 m. No aids, except in adverse terrain or weather, F = normal or near normal gait. * Body weight including the prosthesis. n.a. = not applicable.

**Table 2 sensors-20-06770-t002:** Median and IQR (interquartile range) of the outcome measures for the persons with a lower limb amputation and the elderly control group.

Variable	Persons with A Lower Limb Amputation *n* = 20	Elderly Control Group *n* = 10	*p*-Value
Maximal consecutive steps	141 (60–217)	883 (362–1168)	<0.001
10 MWT (s)	17.4 (10.3–25.5)	9.4 (9.1–10.6)	<0.001

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
