# Peer review of "Maximal Walking Distance in Persons with a Lower Limb Amputation"

_sensors, 2020, doi:10.3390/s20236770_

Round 1
Reviewer 1 Report
The aim of this study was to assess the number of consecutive steps and bouts of walking in patients with a lower limb amputation vis-a-vis their SIGAM mobility grades and compared to age-matched controls.
Overall, this is an excellent paper, well-written with a clear, concise literature review, a straightforward and achievable purpose, and appropriate methods which are properly discussed.
My only strong recommendation is for some minor, but significant, changes in the description of members of the two groups. In my opinion, the current language stigmatizes the target group members and treats an amputation as something 'unhealthy'. First, references to the control group being the 'healthy' group should be edited or removed as necessary. All participants were 'healthy' according to the WHO definition of health. Second, to refer to a participant as an 'amputee' or a 'patient' is not consistent with contemporary language norms. The two groups should be renamed to reflect the purpose of their being grouped and not any descriptions that may be interpreted in a negative way, e.g., "target group" (or similar) and "control group". References to "amputees" could be changed to "target group members" or similar.
Author Response
Please see the attachment.
Kind regards,
Cheriel Hofstad and co-authors

Reviewer 2 Report
The manuscripts attempted to evaluate the maximal walking distance of amputees using measurements from IMUs whereas existing studies using questionnaires may not be reliable. The work in this study could widen the scope of sensors applications in rehabilitation practice to complement traditional evaluation method. Here are some specific comments:
Line 13: important outcome. What outcome? Functional outcome?
Line 14: “it is based on self-report….” What is “it”?
Line 16: The aim described is not precise. The SIGMA in-between amputees and age-matched controls is misleading. Do you mean that age-matched controls do not use SIGMA? Most importantly, the “sensors” are missing (This is the journal, sensors).
Line 18: participated the experiments
Line 21: SIGMA and 10mWT are at different sense (one is an instrument, another is a testing protocol). Both of the methods “were assessed”. This is confusing.
Line 24: poorly correlated or moderately correlated?
Line 25: “there was a discrepancy in 6 amputees” Not understood
Line 26: I think you mean SIGMA does not adequately reflect what amputees do. However your correlation coefficient is quite high. You may need to adjust the tone.
Line 29: “for optimal treatment” is overclaimed based on the research design. Same applies to the conclusion.
Line 48: Why self-reporting is likely to have false positive outcome? Because of bias towards independency? Need to explain more.
Line 49: clinical care that used questionnaires are based on incorrect assumption. This argument is too aggressive. You may fine tune it like may subject to bias or subjectivity.
Line 52: brackets unclosed
Line 59: physically active
Line 60: “None of the studies” is too aggressive. This is a perception without, at least a systematic review. Try to convert into a lack of study
Line 66: physically active
Line 79: Please be reminded that Declaration of Helsinki (2013 version) requires mandatory registration of Human experiments in public registry. You may either put up the registry number or, maybe, just pertained to the ethical approval.
Line 90: You need to explain the type of data the accelerometers are measuring (number of steps). Also, mentioned how do you compromised the data among the three sensors.
Line 92: You may need to have some overview statement generalizing the relationship between 10MWT and two consecutive day testing at the first place. They are confusing.
Line 106: number of steps walked. When? 10MWT or consecutive days?
Line 127: Number of steps is a discrete variable. Using ANOVA is incorrect. You may need to use Chi-square or Fisher Exact test.
Line 131: Addressing significance level of 0.05 required
Line 149: abbreviation of n.a.
Line 198: Figure 3, R and p-value shall be inside the charts
Line 211: do not sufficiently reflect
Line 238: With regards on the reluctancy on the long-distance walking for amputee, there are a couple of relevant reference that shall be cited. Amputees had a poorer joint coordination, and thus may be easier to get fatigue, discomfort and unstable gait. This could be the reasons as well.
Wong, D. W. C., Lam, W. K., Yeung, L. F., & Lee, W. C. (2015). Does long-distance walking improve or deteriorate walking stability of transtibial amputees?. Clinical biomechanics, 30(8), 867-873.
Yeung, L. F., Leung, A. K., Zhang, M., & Lee, W. C. (2013). Effects of long-distance walking on socket-limb interface pressure, tactile sensitivity and subjective perceptions of trans-tibial amputees. Disability and rehabilitation, 35(11), 888-893.
Line 268: The level of amputation is an apparent co-variate that may affect your findings and statistical results requiring discussion.
Author Response

(The authors gave the same response as above.)
